# Using portraits to quantify the changes of generalized social trust in European history: A replication study

**Léonard Guillou**[1], **Lou Safra**[2], **Nicolas Baumard**[1] *

**1** Département d'études Cognitives, Institut Jean Nicod, ENS, EHESS, CNRS, PSL Research University, Paris, France, **2** Sciences Po, CEVIPOF, CNRS, Paris, France

* nbaumard@gmail.com

## Abstract

A portrait is an exercise of impression management: the sitter can choose the impression she or he wants to create in the eyes of others': competence, trustworthiness, dominance, etc. Indirectly, this choice informs us about the qualities that were specifically valued at the time the portrait was created. In a previous paper, we have shown that cues of perceived trustworthiness in portraits increased in time during the modern period in Europe, meaning that people probably granted more importance to be seen as a trustworthy person. More-over, this increase is correlated to economic development. In this study, we aim to replicate this result, using more controlled databases: 1) a newly created database of European head-of-state sovereigns (N = 966, from 1400 to 2020), that is a database of individuals holding the same social position across time and countries, and 2) a database of very high-quality portraits digitized with the same technique, and coming from the same Museum, the Chateau de Versailles database (N = 2,291, from 1483 to 1938). Using mixed effects linear models, we observed in the first dataset that the modeled perceived facial trustworthiness of these sovereigns' faces increased over time ($b = 0.182 \pm 0.04$ s.e.m., $t(201) = 4.40$, $p < 0.001$). On the opposite, no effect of time was detected on the portraits of the Château de Versailles ($b = -0.02 \pm 0.03$ s.e.m., $t(759) = -0.85$, $p > .250$). We conclude by discussing the potential of this new technique to uncover long-term behavioral changes in history, as well as its limitations.

## 1. General introduction

Since the early modern period, European countries have become more and more democratic and more inclusive with increasing power given to parliaments, the enlargement of voting rights, the decrease in state violence [1–6]. The origins of these changes remain elusive though. Using contemporary data and international large scale survey, modernization theory have demonstrated a robust and persistent association between democratic values and economic development [7–9]. According to modernization theory, the values of autonomy, freedom and cooperation increase with economic development. When a large share of the population grows up taking survival for granted, people place growing emphasis on free choice in politics, politi-cal liberties and democratic institutions. Strikingly, these values tend to emerge even under

**Data Availability Statement:** All files are available from the OSF database. https://osf.io/3bf54/.

**Funding:** Dr Nicolas Baumard received funding from Agence Nationale de la Recherche (EUR FrontCog ANR-17-EURE-0017*). The funders had

no role in study design, data collection and analysis, decision to publish, or preparation of the manuscript.The funders had no role in study design, data collection and analysis, decision to publish, or preparation of the manuscript.

**Competing interests:** The authors have declared that no competing interests exist.

authoritarian political regimes. People become economically and physically more secure and more articulate, and want more freedom and more autonomy, whatever the nature of the existing regime [7, 8].

However, large-scale surveys only start in the 1970's, and it is difficult to go back further in times [but see 10, 11]. Moreover, surveys are based on declarative material, which may not accurately reflect people's values and preferences. Scientists must therefore rely on other sources of information to study psychological changes or regularities in the past and to test specific hypotheses about the sources of these changes. In economic history, scientists typically rely on administrative data, such as urbanization rates, literacy rates and average wages [12–14]. These datasets provide valuable quantitative data about the dynamics of past societies, but they are limited in terms of insights into the psychological traits of individuals. On the other hand, historians have analyzed the content of cultural artifacts, be it codes of honor, common and canonical laws, books, portraits, or costumes [15, 16]. These works have provided important insights about the evolution of mentalities and values. However, they are mostly qualitative, which most often preclude the possibility to test hypotheses about the dynamics and the origins of the changes.

Thanks to recent advances in data sciences, it is now possible to move further [17–20]. In this perspective, Baumard et al. [21] relied on the analysis of book descriptions to study the evolution of the prevalence of love stories throughout time and space, showing causal relation between economic development and the rise of love stories. Focusing on the question of social trust, Martins & Baumard [10] relied on the analysis of theatre play to study the evolution of moral norms in Europe between 1550 and the 1900. Similarly, in a recent article, Safra et al., [22] proposed to circumvent this difficulty by using another type of data: portraits. A portrait, just like a photograph or a selfie today, allows one to control the impression one wants to give to others. Someone can choose to look young or old, to be natural or hide physical particularities such as scars. They can also modulate the way our person will be perceived socially. They can choose to smile or not, to appear dominant or not, trustworthy or on the contrary rather competitive and aggressive, etc. Portraits reveal the way the person who ordered the portrait wants to be perceived, and therefore indirectly the values that he or she thinks are important to put forward (youth, beauty, trustworthiness, dominance). But the portraits also reveal the values of the audience of the portrait. Whoever commissions the portrait must indeed adapt to its audience, and does not have the freedom to move too far away from their values. In particular, in societies characterized by a strong reliance on anonymous cooperation (defined as societies with a high level of ''generalized social trust'), appearing as a good cooperation partner, and in particular as a trustworthy one, is central for being included in social interactions and not missing social opportunities. We can thus make the hypothesis that the importance granted of appearing trustworthy notably reflect changes in the importance of cooperation in the society and thus in its level of generalized social trust. Crucially, we cannot directly assess how important it was for one specific individual to appear trustworthy on a portrait or throughout their life, but the analysis of the evolution of multiple portraits produced at different time periods or in different societies can provide information about differences in the importance of being perceived as trustworthy at the global, society level.

Classically, first impressions studies rely on participants' subjective evaluations of unknown faces. However, the use of historical material create obstacles to the use of such procedure as participants' evaluations may be influenced by their previous knowledge on the different time period in which the portraits were painted (participants may have a positive or negative a priori on periods that are often described to the general audience as more or less democratic, open or economically favorable). To counter this issue that can artificially induce an increase

in the modeled perceived facial trustworthiness with time, we relied on the methods presented in Safra et al. [22]. By relying on an automated extraction of face action units, this method allows for the approximation of human-like evaluations of first impressions that are not influenced by historical cues that are present in the portraits, although it is influenced by the other biases that are present in the participants data on which it has been trained. The goal of this method is thus to get measures that would approximate what the subjective evaluations trustworthiness that participants would have on these paintings if they were blind to the historical cues present on the portraits (this measure is referred in this paper to as 'perceived facial trustworthiness'). Using large databases (National Portraits gallery, Web Gallery of Arts), Safra et al. [22] were able to show that cues of perceived trustworthiness in portraits increased during the early modern and modern periods (1450–2000), suggesting that people probably granted more importance to appear as a trustworthy person. Moreover, this increase was correlated to economic development, and predated the democratization of Europe. This last result converges with Modernization theory, as well as with previous works on the long-term evolution of democratic values [10, 11], suggesting that the combination of digital humanities and more classical databases used by economists can provide important insights on the evolution of societies.

However, this study suffers from a number of potential bias. In particular, it could be the case that the observed rise of modeled perceived facial trustworthiness is an artifact that results from the sample of the study and the rising proportion of middle-class portraits. If middle class people are more willing to appear trustworthy, the increasing number in the database over time mechanically would lead to an increase of perceived facial trustworthiness. One way to control for this bias is to restrict the analysis to individuals with similar social status.

To do so, 1,224 portraits of European sovereigns covering a timespan between the 14th century and nowadays were collected online. Although political regimes have changed since the Middle Ages, sovereigns, have very similar social status, being by definition at the top of the social hierarchy. Therefore, any change in the facial representations we could observe could not be attributed to the replacement of a social class by another. Another potential limits of Safra et al. [22] is that the only study on non-English portraits relied on the use of multiple databases and so included portraits that potentially varied in their resolutions. It is thus important to test whether the association between economic development and the evolution of portraits reported in Safra et al. is also present in high-quality and homogeneous digitized paintings from another country. Therefore, in the present study, we conducted on the Chateau de Versailles portrait database the exact same analyses as Safra et al., conducted on the National Portrait Gallery [22].

Following the work of Safra et al. [22], we focused on facial expressions variations through time. To do so, a random-forest algorithm was trained on avatars generated via FaceGen, using the model developed by Oosterhof and Todorov [23]. Our algorithm is based on the activation of the facial muscles (measured by the software OpenFace). This gives the opportunity to free oneself from any intuition or cultural bias related to historical cues, such as clothing, hairstyle, or the background of the portrait [24].

We also studied the underlying determinant of this increase in pro-sociality. In line with the modernization theory and previous work conducted on contemporary data [7–9], Safra et al. (2020) showed an association between economic resources and modeled perceived facial trustworthiness. We aimed at replicating this effect. We thus collected growth domestic product data from the Maddison Project database [25] for each country we had in our dataset. Several mixed effects linear models were computed to assess the effect of GDP on modeled perceived facial trustworthiness and disentangle it from the effect of time.

## 2. Study 1: European sovereigns

### 2.1. Material and methods

**2.1.1. Data collection and pre-processing.** Data collection followed an iterative process due to the difficulty of creating a database large enough to carry analyses similar to Safra et al. [22]. In the end, data collection took four steps. All steps were pre-registered on OSF (https://osf.io/jbs64 and https://osf.io/6hm2c).

We originally planned to use only museum websites because we suspected that portraits were of higher quality on these websites (Step 1). We collected the names of sovereigns using Wikidata API, as well as birth and death dates. These lists were drawn for the following countries: England (or United-Kingdom), Denmark, Spain, France, Poland, Portugal, Holy Roman Empire, Sicilia-Sardinia and Sweden. We then entered each name in the search queries of the major art museums. This first sample was composed of 421 portraits (pre-registered here: https://osf.io/jbs64). The quality checks performed on this sample did not yield significant results: neither the age nor the gender effect was significant (see Statistical Analysis section 2.1.3). In line with our pre-registration, the analyses were stopped.

The absence of gender effect was probably due to the small size of our sample (N = 421) compared the ones from the National Portraits Gallery (N = 1962) and the Web Gallery of Art (N = 4106) used in Safra et al. [22]. Therefore, we decided to perform a power analysis to evaluate the needed sample size to be able to expect potential significant results.

We used the data from Safra et al. [22] to perform a power analysis (keeping in mind this study used only one database, unlike in this study, and focused on many different social categories, again unlike this study). 164 different sample sizes from N = 100 to N = 1490 were tested. Random sub-samples (1000 for each sample size tested) were drawn from the dataset of the National Portraits Gallery of Safra et al. [22]. In order to have samples similar to the one of sovereigns, portraits dated later than 1950 were excluded from the National Portraits Gallery database before drawing our subsamples. Indeed, there were 202 portraits (10%) dated later than 1950 in the dataset of Safra et al. [22] compared to 22 (2.3%) in our dataset. Then we computed a linear model to explain modeled perceived facial trustworthiness with perceived facial dominance, age and gender as control variables and portrait date (time) as variable of interest. Finally, we ploted the p-value of the effect of time in function of the sample size (Fig 1A). From this analysis, using the mean p-values and the 95% CI around them, it appears samples bigger than 750 give p-values lower than 0.05 in more than 96% of the simulations. As we knew that some faces would be excluded because of a bad detection by OpenFace or other exclusion criteria, we decided to collect a total of around 1,200 portraits to secure a sample size of at least 750, and to extend our data collection to new states (Step 2).

To get a higher number of portraits, we added several states: Genoa, Medici, Milan, Montferrat, Savoy, Venice, Luxembourg, Belgium, Netherlands, Scotland, Norway, and Russia (Step 2, preregistered). We collected the names of sovereigns using Wikidata API, as well as birth and death dates. We then entered all these names, one by one, in the search queries of the major art museums. This led us to a total of 643 portraits, which was still under our goal sample size. Therefore, we relaxed our methods and turned to the Wikipedia database (Step 3).

The Wikidata portraits of the sovereigns and their spouses were collected as well as their metadata (painting date, painter's name, technique). Finally, if a new portrait was the same as one already present in our database, then it was eliminated (preregistered https://osf.io/6hm2c). Again, this method did not achieve the expected number. We thus turned to the entire web (Step 4).

We used Google image to look for portraits representing these sovereigns. The same metadata as before were collected. As the number of potential portraits was now much bigger, we

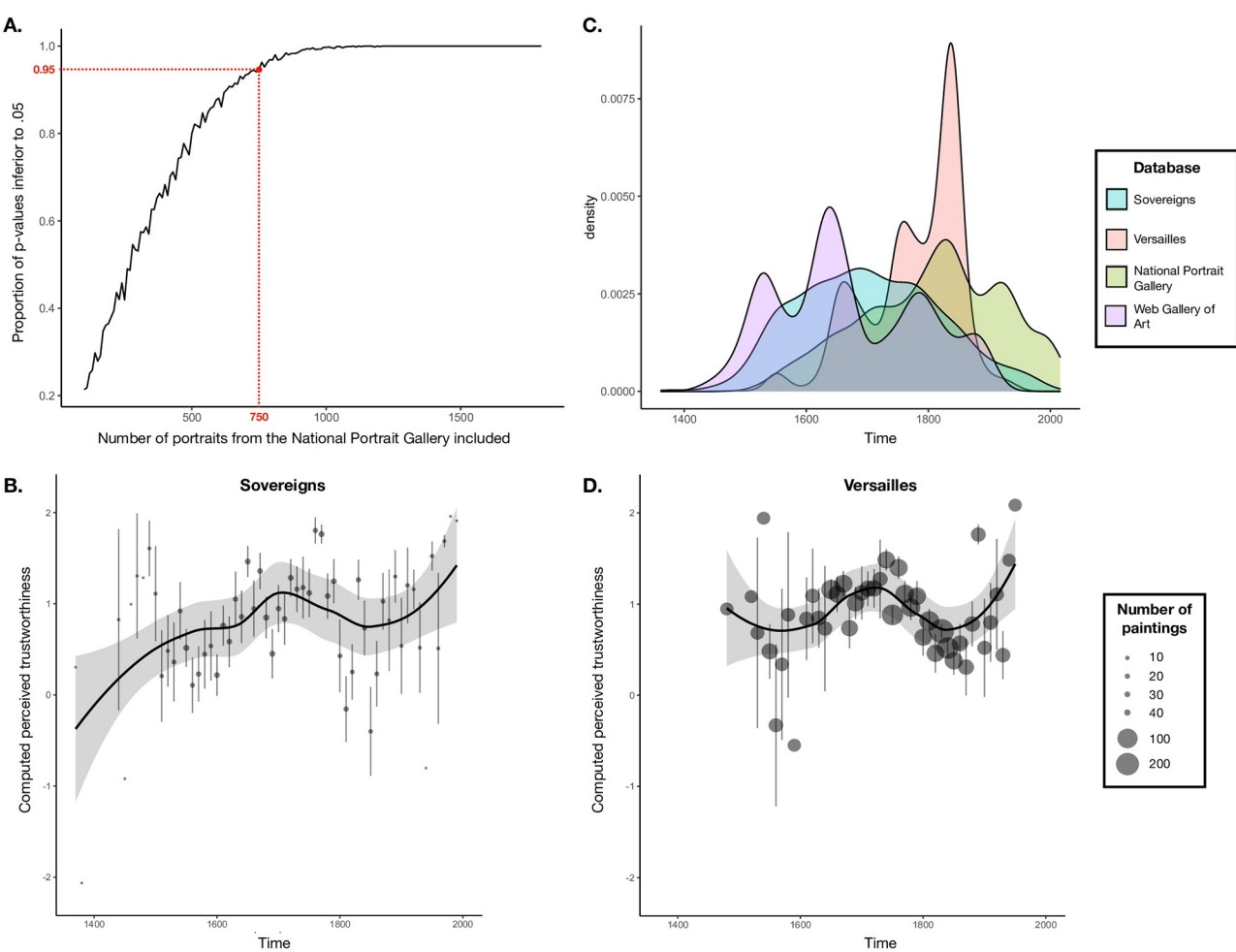

**Fig 1. Evaluation of changes in modeled perceived facial trustworthiness in historical portrait databases.** A. Power analyses conducted on the National Portrait Gallery database from Safra et al. (2020). Models ran on random samples including at least 750 portraits revealed a significant effect of time at least 95% of the times. B. Densities of portraits through time for the datasets analyzed in this study and in Safra et al. (2020). C-D. Variations of mean modeled perceived facial trustworthiness through time in the Sovereigns database (C) and in Versailles database (D). Variations were plotted with a span of 0.7. The grey area is the 95% CI around the mean. Each point corresponds to the average computed perceived trustworthiness on a decade, error bars correspond to the standard deviations. The size of the points increase with the number of portrait analyzed for each decade.

randomly selected 100 and then 150 names in our list of sovereigns to look for more portraits (preregistered). Finally, a total of 581 portraits were collected from steps 3 and 4.

For all portraits (regardless of the collection methodology), profile, low quality and undated ones were not collected. The portraits that were not painted during the lifetime of the sovereign were also removed because we assumed that they would not necessarily reflect the image the sovereign would have liked to display. In the end, we obtained 1224 portraits painted between the 14th and the 20th century (Fig 1B).

**2.1.2. Computation of perceived trustworthiness and perceived dominance levels.**
Level of perceived trustworthiness and perceived dominance of faces were computed based on the facial action units automatically extracted by Open Face 3.1 (https://github.com/TadasBaltrusaitis/OpenFace) and transformed into perceived facial trustworthiness and perceived facial dominance evaluation scores using a random forest algorithm optimized on the avatars controlled for perceived trustworthiness and perceived dominance generated by

FaceGen [23], trained using the exact same method and database as the one reported in Safra et al. (2020).

The OpenFace 3.1 software is not able to process all types of images. Some pictures were excluded because the software did not detect any face on it (due to basic characteristics such as face orientation or insufficient contrast between the background and the face).

Pictures were rated by 3 evaluators (two interns and one the author). Each of the 3 evaluators independently classified the face as accurately detected or not: 0: badly detected; 1: well detected. The three scores were added to attribute a unique detection score to each picture. These detection scores were used as weights for the statistical analyses. Images that were scored three times 0 were therefore excluded from the analyses.

Moreover, when all the data from different searches were combined, we were able to identify and eliminate duplicate images.

The final sample was composed of 329 female faces and 637 male faces painted between 1371 to 1999 (see Table 1). The distribution of the images between the different countries or regions is as follows: Benelux 5 portraits, France 86, Holy Roman Empire 59, Italy 265, Netherlands 63, Poland 35, Portugal 50, Russia 27, Scandinavia 163, Spain 23, United Kingdom 190.

**2.1.3. Statistical analyses.** First of all, to assess the quality of our data and the proper capacity of our algorithm to predict scores of perceived trustworthiness from faces, we checked if we replicated well-known biases classically observed in participants: namely age and gender effects. These biases are of two kinds. First, female faces are on average perceived as more trustworthy and less dominant than male faces [24, 26]. Second, younger faces are on average perceived as more trustworthy and less dominant than older faces [24, 26]. These two effects correspond to our quality checks. If these two quality checks would not have been verified, the study would have stopped there (see pre-registration). Our quality checks were verified using linear models taking modeled perceived facial trustworthiness as dependent variable and either age or gender as independent variable, and controlling for time.

To test our hypothesis that GDP per capita is the variable responsible for the increase in pro-sociality in Europe, we collected GDP data at the country level from Bolt & van Zanden [25]. These data go back to the 16th century for some countries. Unfortunately, several kingdoms do not correspond to a current country (e.g., the Holy Roman Empire) and therefore cannot be assigned GDP data. Eventually, the countries that were assigned GDP data are: UK, France, Italy, Netherlands, Poland, Portugal, Spain, Sweden, Russia, Germany, and Belgium. As GDP data are partial, we assigned to each portrait date the GDP corresponding to the closest available GDP data for its country. However, for 51 portraits, GDP per capita could not be retrieved, analyses including GDP per capita were thus conducted on a subsample of 915 portraits.

Because we compiled data from different countries that are therefore not all equally independent, the main analyses on the evolution of modeled perceived facial trustworthiness were conducted using mixed effects linear models taking country as random effect variable. As pre-registered, we computed the proportion of sovereigns having more than one portrait in our

**Table 1. Descriptive statistics of the paintings included in the analysis (portraits of European sovereigns).**

| Statistic | Mean | St. Dev. | Min | Pctl(25) | Pctl(75) | Max |
|---|---|---|---|---|---|---|
| Facial perceived Trustworthiness | 0.85 | 1.37 | -2.61 | -0.29 | 1.99 | 2.77 |
| Facial perceived Dominance | -0.73 | 1.08 | -2.55 | -1.65 | 0.01 | 2.65 |
| Age (years) | 34.45 | 17.35 | 1.00 | 21.62 | 46.38 | 96.50 |
| Painting Date | 1703 | 113.74 | 1371 | 1612 | 1785 | 1999 |

dataset. As this proportion exceeds 30% (it is 36.0%), we also included sovereigns' Wikidata item as random effect variable. In all our models we also included several controls that are well-known potential confounding variables: age, gender, and modeled perceived facial dominance. Indeed, these three variables are known to correlate with modeled perceived facial trustworthiness [24, 26] and might vary with time. Crucially, because modeled facial dominance was computed in a similar manner to modeled facial trustworthiness, including this variable in our model also allows us to control for the specificity of changes in our dependent variable relative to other facial trait-driven perceptions modeled using our method. Detection scores obtained from the three independent raters were also always added as weights as presented above.

First, we computed a model with modeled perceived facial trustworthiness as dependent variable, GDP as independent variable and controls and weights. To compare the effect of GDP and time, another model was computed, identical to the previous one but including time as an additional predictor. Finally, Bayes factors were also computed. For ease of reading and interpretation, age, time (date of the portrait) and GDP per capita variables were z-scored.

## 2.2. Results

**2.2.1. Quality checks.** As expected, men and older sovereigns were given lower modeled perceived facial trustworthiness scores (effect of gender: $b = -0.36 \pm 0.07$ s.e.m., $t(962) = -5.04$, $p < .001$; effect of age: $b = -0.13 \pm 0.04$ s.e.m., $t(962) = -3.85$, $p < .001$; Table 2).

**2.2.2. Effect of time and economic affluence.** In line with our main hypothesis, mixed model analysis revealed a significant increase in the modeled perceived facial trustworthiness with time ($b = 0.18 \pm 0.04$ s.e.m., $t(330) = 4.42$, $p < .001$; Fig 1C, this effect was also found on the portraits on which a GDP per capita could be retrieved, see Table 3).

**Table 2. Models to assess quality checks on the portraits of European sovereigns.**

| | Dependent variable: | |
| --- | --- | --- |
| | **Perceived Facial Trustworthiness** | |
| | **Quality check on Gender** | **Quality check on Age** |
| Intercept | 0.52*** | 0.28*** |
| | (0.07) | (0.05) |
| Perceived facial Dominance | -0.77*** | -0.77*** |
| | (0.03) | (0.03) |
| Gender Male | -0.36*** | |
| | (0.07) | |
| Age | | -0.13*** |
| | | (0.04) |
| Date | 0.17*** | 0.19*** |
| | (0.03) | (0.03) |
| $R^2$ | 0.42 | 0.41 |

*Notes*: Regression estimates are presented with the standard errors to the mean in parenthesis.

Statistical significance:

*p<0.05;

**p<0.01;

***p<0.001

**Table 3. Models assessing the effect of time and GDP on the portraits of European sovereigns.**

| | *Dependent variable*: | | |
| --- | --- | --- | --- |
| | **Perceived Facial Trustworthiness** | | |
| | **Time-only model** | **GDP-only model** | **Time & GDP model** |
| Intercept | 0.50*** | 0.50*** | 0.50*** |
| | (0.08) | (0.09) | (0.08) |
| Perceived facial Dominance | -0.75*** | -0.75*** | -0.75*** |
| | (0.03) | (0.03) | (0.03) |
| Gender Male | -0.28** | -0.29** | -0.29** |
| | (0.09) | (0.09) | (0.09) |
| Age | -0.12** | -0.11** | -0.12** |
| | (0.04) | (0.04) | (0.06) |
| Date | 0.16*** | | 0.14** |
| | (0.04) | | (0.05) |
| GDP per capita | | 0.13* | 0.06 |
| | | (0.05) | (0.06) |

*Notes*: Analyses were conducted on the 915 portraits on which GDP per capita could be retrieved.

Regression estimates are presented with the standard errors to the mean in parenthesis.

Statistical significance:

*p<0.05;

**p<0.01;

***p<0.001

The model only including an effect of GDP per capita confirmed a positive association between economic development and modeled perceived facial trustworthiness ($b = 0.12 \pm 0.05$ s.e.m., $t$ (176) = 2.32, $p$ = .022; Table 3). However, when time was added in the model, the effect of GDP was not significant anymore ($b = 0.06 \pm 0.06$ s.e.m., $t$(162) = 1.09, $p$ = .279). Bayes Factor analyses revealed that the model with time alone performed better than the model with GDP alone (Bayes Factor = 100) and than the model including both time and GDP (Bayes Factor = 22.22), indicating that time alone better explained the observed variation in facial perceived trustworthiness than GDP.

## 2.3. Discussion

This study aimed at investigating the social importance of being seen as trustworthy, over the last six hundred years. To do so, portraits of European monarchs dated from 1371 to 1999 were collected online. Taking as inputs facial muscles activation measured by OpenFace, we trained an algorithm to derive perceived facial trustworthiness scores. These scores were analyzed in several mixed effects linear models with the country as random effect variable. In line with our hypothesis, modeled perceived facial trustworthiness increased over time. A correlation between GDP per capita and modeled perceived facial trustworthiness was also found. However, this association did not survive to the inclusion of time as a controlling variable, suggesting that this association was rather due to the general increase of GDP per capita throughout the studied period.

Using portraits of sovereigns dating from the 14th to the 20th century, our results replicate the increase of modeled perceived facial trustworthiness with time and economic affluence found by Safra et al. [22]. Although we tried to safeguard from potential problem due to changes in the social class with time by only analyzing portraits of rulers at the head of their

state, it should be noted that the status of sovereigns changed over time: the last one still in place have a lot less power than their ancestries which may have potentially created a bias in our analyses.

Concerning the positive association between the modeled perceived facial trustworthiness and economic affluence, it should be noted that, contrary to what was found by Safra et al. [22], we found that time is a better predictor of the modeled perceived facial trustworthiness than GDP. This difference may be due to the limits of our data: GDP data are partial and for some countries there is no data at all (i.e., Russia not before 1950). The imputation of the dataset by assigning to each portrait the GDP that is the closest to the production of the portrait makes the economic data even more imprecise, hence decreasing our ability to properly test the association between GDP and perceived trustworthiness.

## 3. Study 2: Château de Versailles

### 3.1. Introduction

The goal of our second study was to test if we could replicate Safra et al.'s findings obtained on the National Portrait Gallery [22] on a high-quality digitalized portraits database from another country. To do so, we tried to find a database of good quality portraits. Indeed, previous studies (see https://osf.io/3bf54/) showed that the quality of the portraits is important to derive statistical analyses from them. Thanks to a collaboration with the curator of the Château de Versailles, we were able to access 2,291 portraits, from 1483 to 1938, all digitized in very high resolution.

### 3.2. Material and methods

**3.2.1. Data collection and pre-processing.** Thanks to the generosity of its curator, we were able to access the entire digitized collection of the portraits of the Château de Versailles (N = 2,247 portraits painted between the 15th century and the 20th century, Fig 1B). In addition, a dataset containing information on these portraits (date, subject, painter, technique, etc.) was provided. When possible, we also calculated the age of the subject of the portrait and identify her/his gender. Gender was manually coded based on the names of the sitter and could have been retrieved for 1,399 portraits. However, it should be noted that, contrary to the National Portrait Gallery database, because the portraits do not necessarily represent well-known personalities, several birth and death dates were not available online. In the analyzed dataset, the age could be retrieved only for 836 portraits. Age and gender could be retrieved for 764, which constituted the final database on which the analyses were conducted (Table 4). Concerning the economic data, we collected GDP per capita estimates from the Maddison project [25]. A GDP per capita value was retrieved for all the portraits of the final database. This study was pre-registered (see https://osf.io/cz72y).

**3.2.2. Computation of perceived trustworthiness and perceived dominance levels and statistical analyses.** The exact same analyses were conducted as in the previous study.

**Table 4. Descriptive statistics on the Château de Versailles portraits database.**

| Statistic | Mean | St. Dev. | Min | Pctl(25) | Pctl(75) | Max |
|---|---|---|---|---|---|---|
| Facial perceived Trustworthiness | 0.86 | 1.16 | -2.58 | -0.09 | 1.82 | 2.63 |
| Facial perceived Dominance | -0.54 | 1.07 | -2.50 | -1.48 | 0.19 | 2.62 |
| Age (years) | 42.59 | 18.41 | 0.50 | 29.50 | 59.00 | 90.00 |
| Painting Date | 1766 | 78.83 | 1526 | 1713 | 1819 | 1944 |

Contrary to the previous study, 5 non-White individuals were represented in the portraits of this database, with one of them being the main character of the portraits and the other four being secondary characters. None of these faces were detected which is certainly mainly be due to the profile position of most of these faces and some inherent issues of OpenFace 3.0 to detect faces of non-White individuals. Although this element is an important limit of the used method, in the specific context of our studies it has a very limited impact on the robustness of our results given the very small percentage of non-White faces in our database of historical European portraits. This issue was even absent in the European sovereign database (Study 1) as it contained only White faces. As in the previous study, 3 evaluators (an intern and two of the authors) independently classified the face as accurately detected or not by OpenFace. We used the same quality check as in Study 1.

### 3.3. Results

**3.3.1. Quality checks.** Our two quality checks are verified: male and older faces are rated as lower on modeled perceived facial trustworthiness than female and younger faces (gender: ($b = -0.36 \pm 0.06$ s.e.m., $t(760) = -5.65$, $p < .001$; age: $b = -0.07 \pm 0.03$ s.e.m., $t(760) = -2.31$, $p = .001$; Table 5).

**3.3.2. Effect of time and economic affluence.** No significant effect of time was evidenced in the model ($b = -0.02 \pm 0.03$ s.e.m., $t(759) = -0.85$, $p > .250$; Fig 1D). Similarly, no significant effect of GDP per capita was evidenced whether it was included in the model without controlling for time ($b = -0.01 \pm 0.03$, $t(759) = -0.22$, $p > .250$) or when controlling for time ($b = 0.01 \pm 0.03$, $t(758) = 0.31$, $p > .250$).

### 3.4. Discussion

This study fails to replicate the results from Safra et al. [22]. One explanation could be that the number of portraits having values for gender and age is too low. However, this is quite

**Table 5. Quality checks on the Château de Versailles portraits database.**

| | Dependent variable: | |
| --- | --- | --- |
| | Perceived Facial Trustworthiness | |
| | Quality check on Gender | Quality check on Age |
| Intercept | 0.69*** | 0.42*** |
| | (0.06) | (0.03) |
| Perceived facial Dominance | -0.76*** | -0.80*** |
| | (0.03) | (0.03) |
| Gender Male | -0.36*** | |
| | (0.06) | |
| Age | | -0.07* |
| | | (0.03) |
| Date | -0.03 | -0.02 |
| | (0.03) | (0.03) |
| $R^2$ | 0.59 | 0.58 |

Notes: Regression estimates are presented with the standard errors to the mean in parenthesis.

Statistical significance:

*p<0.05;

**p<0.01;

***p<0.001

unlikely. Indeed, as shown in Study 1, samples with sizes bigger than 750 are very likely to yield p-values for the effect of time lower than 0.05. Moreover, the excellent quality of the digitalization of these portraits might have reduced the noise and therefore allow for the detection of small effects with lower sample sizes.

A second possibility is that the absence of results is due to the inhomogeneous distribution of the portraits over time (see Fig 1). The dataset of the Château de Versailles is less homogeneously distributed over time than the three other datasets where an effect of time has been found (Fig 1B). Indeed, although our dataset included portraits which were produced between 1483 and 1950, the analysis of the distribution of the portraits revealed only very few portraits dated before 1600 or after 1850 (2% and 9% respectively).

## 4. General discussion

These two studies aimed at testing the robustness the results observed in Safra et al.'s paper (2020). Study 1 did replicate the results on the association between time and modeled perceived facial trustworthiness, using a database controlled for the social status of the sitters. However, Study 2 failed to replicate these results and in both databases, the effect of economic development was not significant when time was included as a covariate. This result highlights the limit of the initial findings and of this new methodology. The studied phenomenon (variations in the importance of appearing trustworthy) is probably small and thus requires large enough and controlled samples evenly distributed in time. Such samples are difficult to create in pre-modern societies where paintings were expansive and photograph did not exist. Further, the precision of economic data tends to decrease for more ancient eras [25], making it hard to robustly identify associations across periods and countries. Further, while GDP per capita is amongst the few measures that can be used for studies on long time scales, it has been identified as only providing a partial assessment of societies' economic situation [25]. By contrast, for instance, we studied the evolution of portraits of the French representatives over the period 1910 to 2017 [27]. Because the material is much larger, we were able to collect more than 10.000 pictures, which made the detection of the variations in perceived trustworthiness much easier.

It is also important to note that this methodology does not give access to the psychology of the person posing. It is entirely possible to appear trustworthy in a portrait, without actually being trustworthy. Portraits give us access to how individuals want to be perceived, and the values of their audience that constrain how they can present themselves to others. Crucially, the analysis of individual portraits does not provide information on how these specific individuals on these portraits wanted to be perceived given the influence of individual specificities in self-representation. On the contrary, the analysis of multiple portraits produced in one society that provide information on how these preferences evolved on average in the population. Portraits therefore give us access to what is socially acceptable and socially valued at a given time. However, we must acknowledge that although this method relies on the facial features that have been identified as constituting the 'shared' component of first impressions (in opposition to the idiosyncratic one), it necessarily embeds the cultural biases present in the Western participants populations on which evaluations the first impression models have been trained. Furthermore, the scores extracted by this method, while significant, are only moderately correlated with participants' ratings of trustworthiness. Future studies are needed to further identify the precise limitations of this method.

In addition, not all individuals in a population have equal weight on how individuals should present themselves to others. Social elites obviously carry more weight, especially in more authoritarian eras. The intended audiences of these portraits were likely to be part of the upper-

class (e.g. members of court, wealthy individuals, art patrons, etc.) rather than the middle- and lower-classes who might differ greatly in terms of values. This methodology therefore primarily gives access to the values of the most socially influential part of the population. This does not make the result any less interesting though. It is indeed interesting to note that even in the most privileged parts of the population, those who benefit most from the authoritarian regimes of the time, it is possible to distinguish an evolution towards more democratic values. This can notably contribute to the debate on the causes of democratic revolutions in Europe during the modern period [28–30]., and in particular the role of economic development as an ultimate factor, and psychological changes as a proximate factor in fostering institutional change [10, 29].

Finally, it should be recognized that portraits contain much more information than just the facial features we analyzed in the present study. Pose, costume, the presence of certain objects or animals have been used throughout art history to signal certain statuses, characteristics, and intentions. For example, dogs are known to have been used in official portraiture as symbols of loyalty. Therefore, by relying solely on facial features, our analyses omitted culturally important information contained in portraits. One consequence of this methodological choice is that we cannot rule out the possibility that cooperative intentions were primarily communicated by means other than facial features in a more distant past and that their importance has gradually been replaced by that of facial features in more recent periods, a process that could explain the increase in perceived facial trustworthiness in more recent portraits. However, while the question of the robustness of first impressions of faces across populations and cultures requires further investigation, these other symbols are highly culture-specific (as in the case of parrots; [31]), which precludes their use in longitudinal or comparative studies.

Cognitive sciences and computational social sciences may have a lot to offer to cultural and economic history. They can use traditional material (i.e. portraits, but also texts, costumes, music) in a new way to uncover cultural and potentially psychological traits that are crucial to understand the dynamics of past societies. Using insights from cognitive science studies in artworks are thus one more tool in the emerging field of quantitative psychological history [10, 19, 21, 32, 33]. Obviously, as with any scientific study, these findings must be taken with caution and require replication on different databases and, ideally, using different artifacts to assess their robustness. In the present study, we successfully replicated only one of the results reported by Safra et al. [22] using an independent database in only one of our studies. As with any partial replication, this calls for a critical perspective on the limitations of the methods and the insights they can provide. In particular, it suggests that this methodology is still in its infancy and requires relatively large and well-balanced databases.

## Acknowledgments

We thank three interns: Alexis Garsmeur, Saeyeon Kwon and Marius Mercier who helped us collect and process the portraits of European sovereigns.

For the second study, we thank the curator of the Château de Versailles and an intern: Liza Sarde who helped us collect and process the portraits of the Château de Versailles.

## Author Contributions

**Conceptualization:** Lou Safra, Nicolas Baumard.

**Data curation:** Léonard Guillou.

**Formal analysis:** Léonard Guillou, Lou Safra.

**Project administration:** Nicolas Baumard.

**Writing – original draft:** Léonard Guillou, Lou Safra, Nicolas Baumard.

**Writing – review & editing:** Lou Safra, Nicolas Baumard.

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
