## [Decision Letter · Decision Letter 0]

26 Dec 2022

PONE-D-22-29440Quantifying the changing importance of social trust in the self-presentation of Western European elites throughout history: A replication studyPLOS ONE

Dear Dr. Baumard,

Thank you for submitting your manuscript to PLOS ONE. After careful consideration, we feel that it has merit but does not fully meet PLOS ONE’s publication criteria as it currently stands. Therefore, we invite you to submit a revised version of the manuscript that addresses the points raised during the review process.

 Please submit your revised manuscript by Feb 06 2023 11:59PM. If you will need more time than this to complete your revisions, please reply to this message or contact the journal office at plosone@plos.org. Please include the following items when submitting your revised manuscript:A rebuttal letter that responds to each point raised by the academic editor and reviewer(s). You should upload this letter as a separate file labeled 'Response to Reviewers'.A marked-up copy of your manuscript that highlights changes made to the original version. You should upload this as a separate file labeled 'Revised Manuscript with Track Changes'.An unmarked version of your revised paper without tracked changes. You should upload this as a separate file labeled 'Manuscript'.

We look forward to receiving your revised manuscript.

Kind regards,

Hugh Cowley

Staff Editor

PLOS ONE

Journal Requirements:

4. We note that Figures 2 and 3 in your submission contain copyrighted images. All PLOS content is published under the Creative Commons Attribution License (CC BY 4.0), which means that the manuscript, images, and Supporting Information files will be freely available online, and any third party is permitted to access, download, copy, distribute, and use these materials in any way, even commercially, with proper attribution. For more information, see our copyright guidelines: http://journals.plos.org/plosone/s/licenses-and-copyright.

a. You may seek permission from the original copyright holder of Figures 2 and 3 to publish the content specifically under the CC BY 4.0 license. 

5. We note you have included a table to which you do not refer in the text of your manuscript. Please ensure that you refer to Table 1,2, 4 and 5 in your text; if accepted, production will need this reference to link the reader to the Table.

Additional Editor Comments:

Please note that we have only been able to secure a single reviewer to assess your manuscript. We are issuing a decision on your manuscript at this point to prevent further delays in the evaluation of your manuscript. Please be aware that the editor who handles your revised manuscript might find it necessary to invite additional reviewers to assess this work once the revised manuscript is submitted. However, we will aim to proceed on the basis of this single review if possible. The reviewer has provided comments on the relationship between GDP and perceived trustworthiness, as well as areas for elaboration in the introduction and discussion. Please ensure you address each of the reviewer's major and minor comments, copied below.

Reviewers' comments:

Reviewer's Responses to Questions

**Comments to the Author**

1. Is the manuscript technically sound, and do the data support the conclusions?

Reviewer #1: Partly

2. Has the statistical analysis been performed appropriately and rigorously? 

Reviewer #1: Yes

3. Have the authors made all data underlying the findings in their manuscript fully available?

Reviewer #1: No

4. Is the manuscript presented in an intelligible fashion and written in standard English?

Reviewer #1: Yes

5. Review Comments to the Author

Reviewer #1: The current manuscript presents a replication and extension of Safra et al. (2020) and describes a computational approach to studying the evolution of facial cues to trustworthiness in European portraits over time. In two samples, one broadly focusing on European monarchs/sovereigns and the other on the digitised collection of portraits from the Château de Versailles, the authors find (dataset 1) but then fail to replicate (dataset 2) an effect of time on perceived trustworthiness, indicating that through time there has been an increase in the inclusion of “trustworthy” facial features in portraits of European monarchs.

My impression of the paper is a competent replication and a clear and concise manuscript. The methods and results are clearly communicated and the authors have adhered to the preregistration protocols. Methodological decisions (such as adding new images from Wikipedia, the target sample size, etc.) are clearly communicated and justified. The OSF Project with data and code is not publicly available and so I was not able to look at the raw data directly.

My major concerns with the manuscript as it currently stands relate to: (1) the interpretation of the relationship between GDP and the appearance of trustworthiness in portraits, and (2) the brevity of the introduction and discussion at the expense of important consideration for terminology and limitations.

1. The authors report several times, both in the discussion sections and in the abstract, that the computed perceived trustworthiness of portraits was predicted by the GDP per capita. However, from what I can tell this was only true when this was the only fixed factor, and the effect of GDP did not survive when controlling for time. This could be explained if GDP correlates with time (i.e. European countries’ GDP tend to increase over time, and so the GDP is predictive of portraits’ appearance only in so far as it is also related to time). In any case, the role of GDP per capita in explaining the appearance of the selected portraits is oversold given that this effect does not hold when time is controlled.

2. I found both the introduction and discussion to be quite brief. The introduction makes a good case for using modern computational approaches to study predictions of modernisation theory but does not link this at all to trust or appearances of trustworthiness and why these are useful to study. I understand that modernisation theory argues that with economic development comes a greater adoption of democratic institutions, but the link with social attitudes such as trust (and leaders’ consequent increased need to appear trustworthy to their subjects) is taken as read. A clearer link and description would be particularly relevant as it would allow the authors to establish their terminology early on—trust and trustworthiness are concepts that can have different connotations in different literatures, and I am aware that imprecision on this matter was an issue for the Safra 2020 paper. It is important in the introduction to spell out what the phenomenon of interest is (self-presentation of European sovereigns and the conveyance of certain social impressions), the proposed hypothesis being tested (that this drive for self-presenting as trustworthy was a result of changing societal pressure and would increase as a function of their society’s economic development), and how this is approached (an algorithm that aims to recreate human-like judgements of trustworthiness from facial features, subject to similar cultural and racial biases as the human data on which it was trained).

Similarly, the general discussion is extremely short and while it does mention a few limitations these are mostly about accessibility of image databases rather than limits of what can be learned from this approach more generally. For example, the authors argue that these results show that cognitive and computational social sciences can offer insights to cultural and economic history by uncovering cultural and psychological traits, but it is not clear what kind of psychological trait the authors mean. Signals of trustworthiness encoded in features of the paintings (e.g. facial expression, head position) may let us infer some of the expectations and attitudes of contemporary viewers and the intentions (tacit or explicit) of the artist and subject. However, algorithmic detection of facial features cannot tell us about the actual trustworthiness of these individuals, which would also be a psychological trait.

In addition, related to the line on p.4, “we can assume that observing the representation of rulers will give at least a partial information on their people’s preferences” – it is also worth considering who “their people” are, because not all viewers’ perceptions are equal, and these portraits would likely have an intended audience (e.g. members of court, wealthy and/or upper class individuals, art patrons, etc. rather than the rank and file who might differ greatly in terms of cultural, racial, and economic background).

Some more minor points follow:

I am unfamiliar with the Madison project and so perhaps this is a naïve question, but how was GDP calculated for city-states that were part of what is today a single country? For example, were the Genova, Medici, Milan, and Venice states differentiated in terms of GDP or did they all fall under “Italy”? If so, how was the GDP for countries such as Italy or Germany calculated prior to the 19th century when these countries did not exist as single states?

Given that the portraits were sampled from European monarchs and sovereigns, I presume that they were mostly (or entirely) White faces, but this should be reported, particularly if this affected the automated facial action unit extraction (e.g. if the algorithm found it harder to identify non-White faces).

Both datasets appear to include portraits of infants or young children (min age 1.00 in Table 1, 0.50 in Table 4). I am curious why a lower age cutoff was not applied given the authors’ stated hypothesis that these portraits reflect a cultivated image that the individuals presented would like to display (cf. p.9). It is unlikely that a 1-year old is concerned about appearing more trustworthy to their subjects.

The presentation of statistics in the tables could be much clearer. Table legends such as “Descriptive statistics” and column headings of (1) and (2) without intelligible or meaningful labels make it quite difficult to keep track of which analysis is being presented (for example, the quality checks were described on p.10 and their results were reported but no mention was made of Table 2 on p.12).

Related to my second major point, in the Discussion of Experiment 1 on p.14 the authors refer to “social trust” – this is a point where it would be beneficial to have explicit, clarified terminology that establishes exactly what is meant by terms like trust, trustworthiness, perceived trustworthiness, displayed trustworthiness, etc.

6. PLOS authors have the option to publish the peer review history of their article (what does this mean?). If published, this will include your full peer review and any attached files.

Reviewer #1: No

---

## [Author Response · Author response to Decision Letter 0]

25 May 2023

Editor's comment

Response: Corrected for Fig 1.

Response: We have not been able to find the Financial Disclosure section. The grant is EUR FrontCog ANR-17-EURE-0017*. This reference is the same in the paper and editorial manager.

Response: Done

4. We note that Figures 2 and 3 in your submission contain copyrighted images. All PLOS content is published under the Creative Commons Attribution License (CC BY 4.0), which means that the manuscript, images, and Supporting Information files will be freely available online, and any third party is permitted to access, download, copy, distribute, and use these materials in any way, even commercially, with proper attribution. For more information, see our copyright guidelines: http://journals.plos.org/plosone/s/licenses-and-copyright.

Response: We have removed the figures from the submission.

5. We note you have included a table to which you do not refer in the text of your manuscript. Please ensure that you refer to Table 1,2, 4 and 5 in your text; if accepted, production will need this reference to link the reader to the Table.

Response: We have added the citations to the text. 

Review Comments to the Author

Reviewer #1: The current manuscript presents a replication and extension of Safra et al. (2020) and describes a computational approach to studying the evolution of facial cues to trustworthiness in European portraits over time. In two samples, one broadly focusing on European monarchs/sovereigns and the other on the digitised collection of portraits from the Château de Versailles, the authors find (dataset 1) but then fail to replicate (dataset 2) an effect of time on perceived trustworthiness, indicating that through time there has been an increase in the inclusion of “trustworthy” facial features in portraits of European monarchs.

My impression of the paper is a competent replication and a clear and concise manuscript. The methods and results are clearly communicated and the authors have adhered to the preregistration protocols. Methodological decisions (such as adding new images from Wikipedia, the target sample size, etc.) are clearly communicated and justified. The OSF Project with data and code is not publicly available and so I was not able to look at the raw data directly.

Response: We would like to thank Reviewer 1 for their positive and constructive comments on our manuscript. We created an anonymized view-only link to our OSF Project so data and scripts are openly accessible to the reviewer https://osf.io/buv7g/?view_only=6193f19fa053485b8db95bf8267c11f3 .

My major concerns with the manuscript as it currently stands relate to: (1) the interpretation of the relationship between GDP and the appearance of trustworthiness in portraits, and (2) the brevity of the introduction and discussion at the expense of important consideration for terminology and limitations.

1. The authors report several times, both in the discussion sections and in the abstract, that the computed perceived trustworthiness of portraits was predicted by the GDP per capita. However, from what I can tell this was only true when this was the only fixed factor, and the effect of GDP did not survive when controlling for time. This could be explained if GDP correlates with time (i.e. European countries’ GDP tend to increase over time, and so the GDP is predictive of portraits’ appearance only in so far as it is also related to time). In any case, the role of GDP per capita in explaining the appearance of the selected portraits is oversold given that this effect does not hold when time is controlled.

Response: We agree. We have removed this result from the abstract and included it in the general discussion together with a sentence discussing potential limits of our economic measure. In the rest of the article, we were very clear that the effect of GDP per capita does not survive when controlling for time. 

Abstract:

Using mixed effects linear models, we observed in the first dataset that the perceived facial trustworthiness of these sovereigns’ faces increased over time (b = 0.182 ± 0.04 s.e.m., t(201) = 4.40, p < 0.001).

General Discussion

However, Study 2 failed to replicate these results and in both databases, the effect of economic development was not significant when time was included as a covariate. 

General Discussion

Further, the precision of economic data tends to decrease for more ancient eras, making it hard to robustly identify associations across periods and countries. Further, while GDP per capita is amongst the few measures that can be used for studies on long time scales, it has been identified as only providing a partial assessment of societies’ economic situation. 

Discussion Study 1:

Concerning the positive association between perceived facial trustworthiness and economic affluence, it should be noted that, contrary to what was found by Safra et al. (2020), we found that time is a better predictor of perceived facial trustworthiness than GDP. This difference may be due to the limits of our data: GDP data are partial and for some countries there is no data at all (i.e., Russia not before 1950).

Discussion Study 2:

No significant effect of time was evidenced in the model (b = -0.02 ± 0.03 s.e.m., t(759) = -0.85, p > .250; Figure 1D). Similarly, no significant effect of GDP per capita was evidenced whether it was included in the model without controlling for time (b = -0.01 ± 0.03, t(759) = -0.22, p > .250) or when controlling for time (b = 0.01 ± 0.03, t(758) = 0.31, p > .250; Table 5).

2. I found both the introduction and discussion to be quite brief. The introduction makes a good case for using modern computational approaches to study predictions of modernisation theory but does not link this at all to trust or appearances of trustworthiness and why these are useful to study. I understand that modernisation theory argues that with economic development comes a greater adoption of democratic institutions, but the link with social attitudes such as trust (and leaders’ consequent increased need to appear trustworthy to their subjects) is taken as read. A clearer link and description would be particularly relevant as it would allow the authors to establish their terminology early on—trust and trustworthiness are concepts that can have different connotations in different literatures, and I am aware that imprecision on this matter was an issue for the Safra 2020 paper. It is important in the introduction to spell out what the phenomenon of interest is (self-presentation of European sovereigns and the conveyance of certain social impressions), the proposed hypothesis being tested (that this drive for self-presenting as trustworthy was a result of changing societal pressure and would increase as a function of their society’s economic development), and how this is approached (an algorithm that aims to recreate human-like judgements of trustworthiness from facial features, subject to similar cultural and racial biases as the human data on which it was trained).

Response: We totally agree and we thank the reviewer for that. Indeed, it is very important to explicit all the links from portraits to democratic values as well as the rationale and the limits of our methodological choices. we have added a new paragraph. We hope that this is clearer.

Introduction:

However, large-scale surveys only start in the 1970's, and it is difficult to go back further in times (but see Martins & Baumard, 2020; Ruck et al., 2019). Moreover, surveys are based on declarative material, which may not accurately reflect people's values and preferences. In a recent article, Safra et al., (Safra et al., 2020) proposed to circumvent this difficulty by using another type of data: portraits. A portrait, just like a photograph or a selfie today, allows one to control the impression one wants to give to others. You can choose to look young or old, to be natural or to correct facial imperfections. We can also modulate the way our person will be perceived socially. We can choose to smile or not, to appear dominant or not, trustworthy or on the contrary rather competitive and aggressive, etc. The portraits reveal the way the person who ordered the portrait wants to be perceived, and therefore indirectly the values that he or she thinks are important to put forward (youth, beauty, trustworthiness, dominance). But the portraits also reveal the values of the audience of the portrait. Whoever commissions the portrait must indeed adapt to his audience, and does not have the freedom to move too far away from their values. In particular, in societies characterized by a strong reliance on anonymous cooperation (defined as societies with a high level of ‘'generalized social trust’), appearing as a good cooperation partner, and in particular as a trustworthy one, is central for being included in social interactions and not missing social opportunities. We can thus make the hypothesis that the importance granted of appearing trustworthy notably reflect changes in the importance of cooperation in the society and thus in its level of generalized social trust. Crucially, we cannot directly assess how important it was for one specific individual to appear trustworthy on a portrait or throughout their life, but the analysis of the evolution of multiple portraits produced at different time periods or in different societies can provide information about differences in the importance of being perceived as trustworthy at the global, society level. 

Classically, first impressions studies rely on participants’ subjective evaluations of unknown faces. However, the use of historical material create obstacles to the use of such procedure as participants’ evaluations may be influenced by their previous knowledge on the different time period in which the portraits were painted (participants may have a positive or negative a priori on periods that are often described to the general audience as more or less democratic, open or economically favorable). To counter this issue that can artificially induce an increase in the perceived facial trustworthiness with time, we relied on the methods presented in Safra et al. (2020). By relying on an automated extraction of face action units, this method allows for the extraction of human-like evaluations of first impressions that are not influenced by historical cues that are present in the portraits, although it is influenced by the other biases that are present in the participants data on which it has been trained. The goal of this method is thus to get measures that would approximate what the subjective evaluations trustworthiness that participants would have on these paintings if they were blind to the historical cues present on the portraits (this measure is referred in this paper to as ‘perceived facial trustworthiness’). 

General discussion:

However, we must acknowledge that although this method relies on the facial features that have been identified as constituting the ‘shared’ component of first impressions (in opposition to the idiosyncratic one), it necessarily embeds the cultural biases present in the Western participants populations on which evaluations the first impression models have been trained.

Similarly, the general discussion is extremely short and while it does mention a few limitations these are mostly about accessibility of image databases rather than limits of what can be learned from this approach more generally. For example, the authors argue that these results show that cognitive and computational social sciences can offer insights to cultural and economic history by uncovering cultural and psychological traits, but it is not clear what kind of psychological trait the authors mean. Signals of trustworthiness encoded in features of the paintings (e.g. facial expression, head position) may let us infer some of the expectations and attitudes of contemporary viewers and the intentions (tacit or explicit) of the artist and subject. However, algorithmic detection of facial features cannot tell us about the actual trustworthiness of these individuals, which would also be a psychological trait.

Response: We totally agree with the reviewer. We have added a paragraph to clarify and explain this limit. 

General Discussion:

It is also important to note that this methodology does not give access to the psychology of the person posing. It is entirely possible to appear trustworthy in a portrait, without actually being trustworthy. Portraits give us access to how individuals want to be perceived, and the values of their audience that constrain how they can present themselves to others. Crucially, the analysis of individual portraits does not provide information on how these specific individuals on these portraits wanted to be perceived given the influence of individual specificities in self-representation. On the contrary, the analysis of multiple portraits produced in one society that provide information on how these preferences evolved on average in the population. Portraits therefore give us access to what is socially acceptable and socially valued at a given time. 

In addition, related to the line on p.4, “we can assume that observing the representation of rulers will give at least a partial information on their people’s preferences” – it is also worth considering who “their people” are, because not all viewers’ perceptions are equal, and these portraits would likely have an intended audience (e.g. members of court, wealthy and/or upper class individuals, art patrons, etc. rather than the rank and file who might differ greatly in terms of cultural, racial, and economic background).

Response: We totally agree with the reviewer. Thanks for pointing that out. We have added a paragraph in the final discussion:

General Discussion:

In addition, not all individuals in a population have equal weight on how individuals should present themselves to others. Social elites obviously carry more weight, especially in more authoritarian eras. The intended audiences of these portraits were likely to be part of the upper-class (e.g. members of court, wealthy individuals, art patrons, etc.) rather than the middle- and lower-classes who might differ greatly in terms of values. This methodology therefore primarily gives access to the values of the most socially influential part of the population. This does not make the result any less interesting though. It is indeed interesting to note that even in the most privileged parts of the population, those who benefit most from the authoritarian regimes of the time, it is possible to distinguish an evolution towards more democratic values. This can notably contribute to the debate on the causes of democratic revolutions in Europe during the modern period (Acemoglu & Robinson, 2012; Baumard, 2018; Fukuyama, 2011).

Some more minor points follow:

I am unfamiliar with the Madison project and so perhaps this is a naïve question, but how was GDP calculated for city-states that were part of what is today a single country? For example, were the Genova, Medici, Milan, and Venice states differentiated in terms of GDP or did they all fall under “Italy”? If so, how was the GDP for countries such as Italy or Germany calculated prior to the 19th century when these countries did not exist as single states?

Response: Yes, they all fall under "Italy". Although they were independent (just as many German states), the GDP of Italy is computed at the national level today. We assume here that they had similar level of economic development. This is a reasonable assumption given that Italian city states had similar economic trajectories during the studied period, as explained in this recent article on the evolution of GDP per capita in Florence:

"In fact, there is little doubt that the XVII century was a period of economic decline for Italy, and especially for the Centre-North, at least for what concerns industry and services; unlike the XVI one. Cipolla made this point as early as in 1952: at the beginning of the XVII century, the Centre-North of Italy was still one of the most advanced industrial areas of Europe, with an «exceptionally high standard of living»; by the end of that century, «Italy had become an economically backward and depressed area; its industrial structure had almost collapsed, its population was too high for its resources, its economy had become primarily agricultural». According to Cipolla, the decline occurred between 1600 and 1670. It was in these seven decades that, first of all, «the industrial structure of Italy collapsed» [Cipolla, 1952, p. 178]: it was true for the woollen industry in Venice, Florence, Milan or other towns of Lombardy; for the silk industry in Genoa or Pavia, as well as for a number of other industries, in several cities throughout the Centre- North; some industrial delocalization did take place, in favour of the countryside, but this was by far not enough to compensate for the fall in urban manufactures.2 At the same time, also the maritime and banking services fell apart. The relative prosperity of Italy depended precisely on exporting textiles (mainly wool and silk) and international services such as banking operations and maritime transports: in both, the Italians were replaced during the XVII century by English, Dutch and even French competitors, all over Europe and the Mediterranean basin. "

van Zanden, J. L., & Felice, E. (2017). Benchmarking the middle ages. XV century Tuscany in European perspective. Centre for Global Economic History Working Paper Series, (81)

Given that the portraits were sampled from European monarchs and sovereigns, I presume that they were mostly (or entirely) White faces, but this should be reported, particularly if this affected the automated facial action unit extraction (e.g. if the algorithm found it harder to identify non-White faces).

Response: We agree that although our datasets contained very few non-White faces (5 individuals over more than faces 2,500 of the Château de Versailles portraits database, with only one portrait over the 2,247 portraits of the database with a non-White individual being the main character; and no face in the European Sovereign database), it is an important element to report in the manuscript. In the revised manuscript we clearly add this information in the Methods section of the study on the Château de Versailles portraits database.

Section 3.2.2. Computation of perceived trustworthiness and perceived dominance levels and statistical analyses

‘Contrary to the previous study, 5 non-White individuals were represented in the portraits of this database, with one of them being the main character of the portraits and the other four being secondary characters. None of these faces were detected which is certainly mainly be due to the profile position of most of these faces and some inherent issues of OpenFace 3.0 to detect faces of non-White individuals. Although this element is an important limit of the used method, in the specific context of our studies it has a very limited impact on the robustness of our results given the very small percentage of non-White faces in our database of historical European portraits.’

Both datasets appear to include portraits of infants or young children (min age 1.00 in Table 1, 0.50 in Table 4). I am curious why a lower age cutoff was not applied given the authors’ stated hypothesis that these portraits reflect a cultivated image that the individuals presented would like to display (cf. p.9). It is unlikely that a 1-year old is concerned about appearing more trustworthy to their subjects.

Response: This is true. Our idea was that, the entourage of the sovereign seek to give a certain impression with the portrait of the young sovereign even at very young age.

The presentation of statistics in the tables could be much clearer. Table legends such as “Descriptive statistics” and column headings of (1) and (2) without intelligible or meaningful labels make it quite difficult to keep track of which analysis is being presented (for example, the quality checks were described on p.10 and their results were reported but no mention was made of Table 2 on p.12).

Response: We agree with Reviewer 1 that the description of the statistics in the tables would benefit to be clearer. In the revised version of the manuscript, we use clearer titles for the tables and added labels for each of the models as well a more precise description of the presented statistics. We also made sure that all the tables were appropriately cited in the main text.

Related to my second major point, in the Discussion of Experiment 1 on p.14 the authors refer to “social trust” – this is a point where it would be beneficial to have explicit, clarified terminology that establishes exactly what is meant by terms like trust, trustworthiness, perceived trustworthiness, displayed trustworthiness, etc.

Response: This is true and we added a definition of the two key terms (‘generalized social trust’ and ‘perceived facial trustworthiness’) in the introduction of the revised version of the manuscript. 

Introduction:

In particular, in societies characterized by a strong reliance on anonymous cooperation (defined as societies with a high level of ‘'generalized social trust’), appearing as a good cooperation partner, and in particular as a trustworthy one, is central for being included in social interactions and not missing social opportunities.

Introduction:

The goal of this method is thus to get measures that would approximate what the subjective evaluations trustworthiness that participants would have on these paintings if they were blind to the historical cues present on the portraits (this measure is referred in this paper to as ‘perceived facial trustworthiness’). 

In addition, we were careful in using a consistent terminology all over the article, referring to "perceived facial trustworthiness" when we talk about what we measure, to "the importance of appearing trustworthy" when we talk about the intention of the patron, and to "the importance of generalized social trust" when we talk about what we infer from our measure. We have been careful not to use "displayed trustworthiness

---

## [Decision Letter · Decision Letter 1]

3 Jul 2023

PONE-D-22-29440R1Quantifying the changing importance of social trust in the self-presentation of Western European elites throughout history: A replication studyPLOS ONE

Dear Dr. Baumard,

Thank you for submitting your manuscript to PLOS ONE. After careful consideration, we feel that it has merit but does not fully meet PLOS ONE’s publication criteria as it currently stands. Therefore, we invite you to submit a revised version of the manuscript that addresses the points raised during the review process.

We look forward to receiving your revised manuscript.

Kind regards,

Anastassia Zabrodskaja, Ph.D.

Academic Editor

PLOS ONE

**Additional Editor Comments:**

Please follow the reviewers' comments and make corrections accordingly.

Reviewers' comments:

Reviewer's Responses to Questions

**Comments to the Author**

1. If the authors have adequately addressed your comments raised in a previous round of review and you feel that this manuscript is now acceptable for publication, you may indicate that here to bypass the “Comments to the Author” section, enter your conflict of interest statement in the “Confidential to Editor” section, and submit your "Accept" recommendation.

Reviewer #1: (No Response)

Reviewer #2: (No Response)

2. Is the manuscript technically sound, and do the data support the conclusions?

Reviewer #1: Yes

Reviewer #2: Partly

3. Has the statistical analysis been performed appropriately and rigorously? 

Reviewer #1: Yes

Reviewer #2: Yes

4. Have the authors made all data underlying the findings in their manuscript fully available?

Reviewer #1: Yes

Reviewer #2: No

5. Is the manuscript presented in an intelligible fashion and written in standard English?

Reviewer #1: Yes

Reviewer #2: Yes

6. Review Comments to the Author

Reviewer #1: The authors have addressed the substantial comments from the previous review. I think this has greatly improved the manuscript - particularly in terms of attenuating any overly bold claims about the ability of the methodology to detect actual trustworthiness (as a personality trait). However, I do have some outstanding comments, some driven by the revisions to the manuscript and others inspired by a second examination of the paper. Once again, my substantive comments are that there should be more explicit acknowledgement of the conceptual limitations of this approach and of the use of algorithms to model human judgements.

I thank the authors for providing a view-only link to the OSF project. I notice however that there are some files referenced in the scripts that are not available on the project, making it impossible to recreate the power analysis reported. I also notice that the scripts as available do not directly replicate the analysis reported in the manuscript - the model coefficients in Table 2, for example, are as reported for dominance, gender, and age, but with different coefficients for the intercept and effect of date. I suspect this may have to do with the default format of the predictor across different versions of R but I mention it so that the authors can double check

The authors acknowledge at one point that although GDP is a useful measure for tracking long-term changes in across centuries and countries, it is only a partial assessment and as you get earlier in history the data become more sparse and uncertain. I think it is also worth acknowledging that the same could also be true for trustworthiness. While there is increasing evidence that first impressions of faces based on physical features are robust across naturalistic images and across cultures (see https://doi.org/0.1177/0146167217744194), it is still a strong assumption--and a central one to the authors' work--that such physiognomy-driven impressions are a human universal and always have been. While I do not think this assumption is unwarranted, it is worth acknowledging and considering alternative hypotheses (e.g. the reason that the prevalence of 'trustworthy' facial features and expressions increases over time is due to increased reliance on physiognomic cues to trust rather than, say, postural or setting cues such as praying).

It would also be worth considering the limitations of automated feature extraction. I understand that the Open Face algorithm is optimised on computer-generated avatars that are specifically designed to vary in perceived trustworthiness and perceived dominance. The use of this algorithm to estimate human judgements of real-face photographs has been validated, but its application to generated faces presents new questions and challenges. Do lower ratings of perceived trustworthiness in earlier portraits reflect a lower motivation to be seen as trustworthy, or do they reflect increased difficulty for the algorithm in extracting and identifying the relevant facial features? Is an increase in perceived trustworthiness over time related to growing democratic norms or is it just a side-effect of a cultural shift towards realism and dynamic expression in portraiture providing more information to the algorithm?

I am not sure how the authors could address this empirically without substantial extra work to validate the performance of the algorithm on non-natural faces, but one approach could be to examine whether algorithmically computed ratings of dominance (or another social judgement) also increase over time or whether this change is specific to perceived trustworthiness. If both ratings increase over time this could indicate that the results are driven by an increase in the amount of information available to the algorithm rather than a specific increase in motivation to appear trustworthy. I see that the authors included dominance as a factor in the models predicting trustworthiness but this is not fully justified beyond the fact that dominance is a known confound. This should be explained further, in any case.

Some more minor comments:

There are a number of typographical and grammatical mistakes that the authors will want to check because PLOS does not copyedit manuscripts before they go to print and relies on reviewers and authors to correct any typographical errors themselves. Some of these are minor, such as the overuse of the definite article where it is not necessary (e.g. the first 'the' in the following sentence: "The portraits reveal the way the person who ordered the portrait wants to be perceived..."), but some could disrupt reader comprehension. I identify some of these here and there but the authors should check through carefully for more

p.2: In the second paragraph of the general introduction, in the new addition, the voice changes from the second person "You" to first person "We". I usually find it preferable to reserve "We" for talking specifically about decisions and actions made by the authors.

p.2: "You can choose to look young or old, to be natural or to correct facial imperfections" - I understand what the authors mean here, but "imperfections" is perhaps a bit strong and may be perceived as inappropriate or even offensive. I would clarify that such "corrections" involve editing or removing facial features that do not conform to contemporary aesthetic standards

p.4: "Another potential limits of Safra et al... is that the only study on non-English portraits relied on the use of multiple databases and thus potentially varying in their resolutions" - change to "Another potential limit... relied on the use of multiple database and so included portraits that potentially varied in their resolution"

p.4: "This gives the opportunity to free oneself from any intuition or cultural bias related to clothing, hairstyle, or the background of the portrait" - although it is still subject to other cultural biases determine by the constitution of the data on which it was trained

p.4: "Our candidate is economic resources." - the details and predictions of the modernisation theory could be reiterated here to clarify why economic factors are expected to relate to self-presentation

p.5: "The quality checks performed on this sample..." I recommend adding a reference to the later section (e.g. "see Statistical Analysis section") so that readers know this is not something they need to look for in the preregistration documents

p.5: "In order to have samples similar to the one of sovereigns, portraits dated later than 1950 were excluded before drawing our subsamples." - this is not clear and I do not understand what the authors did here, considering that it appears there were post-1950 portraits in both datasets?

p.7: While the authors do mention race when describing the second dataset, it would also be helpful to mention that all faces were White in the description of the final sample

p.13: "A correlation between GDP per capita and perceived facial trustworthiness was also found" - also in the first paragraph of the discussion, the authors should take care to remind readers that this relationship did not survive when time was also included in the model. Also in the following paragraph, "our results replicate the increase of perceived facial trustworthiness with time and economic affluence found by Safra et al" - but the current results do not replicate that relationship, and the only time such a relationship is observed it appears to be due to a confound

p.17: The description of the methods and results gets quite confusing with so many different things being compared. For example, what was the final sample size subjected to analysis in section 3.3.3?

p.17: "the analysis of the distribution of the portraits reveals only very few portraits dated before 1600 or after 1850 (Kolmogorov-Smirnov test..." - the KS-test does not back up the statement it follows, as no reference is made to the normality of the distribution

p.17: "Study 1 did replicate the results..." - Once again, this implies that the current findings support a relationship between perceived trustworthiness and yGDP, which is not the case

p.18: "This result [failure to replicate] highlights the limits of this new methodology" - I would advise rewording this as by my personal reading, this suggests that the authors are undermining their own methodology because it does not support the theory whose predictions they set out to test. This gives an impression of 'moving the goalposts' rather than considering what implications this failure to replicate have for the theory itself. I do not think that this is what the authors intended, and so they should clarify

p.19: "This can notably contribute to the debate on the causes of democratic revolutions in Europe during the modern period" - I am unfamiliar with this debate or the literature, but this seems like quite a strong claim. It might be helpful to clarify what, exactly, this contribution might look like

p.19: "Cognitive studies of social impressions in the arts are thus one more a new tools..." - this is poorly worded and I am not sure what exactly the authors are trying to say

Reviewer #2: 1. The study presents the results of original research.

Yes.

2. Results reported have not been published elsewhere.

A Google Scholar search did not turn up any evidence that this work has been previously published.

3. Experiments, statistics, and other analyses are performed to a high technical standard and are described in sufficient detail.

- For the random forest model, it's unclear whether the authors are using one of their models from their previous work, or have developed a new model for the current paper.

- Table 5 shows the gender/age quality check regression results, but the reference in the text (§3.3.3, page 56 of the review packet) suggests it should show the analytic regression results (effects for time and GDP).

4. Conclusions are presented in an appropriate fashion and are supported by the data.

To my mind, the crucial question for both the current paper and Safra et al. (2020) is whether and to what extent the authors' method measures trustworthiness. Insofar as they are not measuring trustworthiness, their conclusions about trustworthiness cannot be supported.

Following their previous work, the authors propose that the "perceived trustworthiness" of figures in historical portraits can be measured using a random forest (RF) model trained on facial action units and the first principal component of an exploratory PCA of descriptions of synthetic faces conducted by Oosteroff and Todorov (2008). This measurement technique relies on several crucial assumptions:

1. There is a universal/culturally independent construct of trustworthiness

2. Oosteroff and Todorov's (2008) first principal component operationalizes this construct (rather than age, gender, some uninterpretable composite, etc.)

3. Accuracy of the RF model at predicting ground truth/human assessments of trustworthiness

4. Extrapolation from the early 21st century WEIRD samples used by Oosteroff and Todorov (2008) and others to contemporaneous viewers of the historical portraits

5. That trustworthiness of portrait subjects was communicated primarily by facial features, rather than other features of the subject such as pose and dress

6. And rather than other features of the portrait, such as the presence of other characters, inanimate objects, or background and their relationship to the subject

Assumption 3 was acknowledged as a weakness in Safra et al. (2020): they reported this accuracy as only r = 0.22. Assumption 4 was also identified as a key assumption ("the assumption that facial cues that are used as cues to assess perceived trustworthiness are shared across time"), with the authors writing that "further work is needed to fully test this assumption." Criticisms related to the other assumptions were raised in the peer review of Safra et al. (2020) (made available by the publisher at https://static-content.springer.com/esm/art%3A10.1038%2Fs41467-020-18566-7/MediaObjects/41467_2020_18566_MOESM3_ESM.pdf) and public discussion after Safra et al. (2020) was published.

In this light, I would have expected follow-up work on Safra et al. (2020) to systematically study (and, hopefully, empirically confirm) these assumptions, conceptually and psychometrically validating the authors' technique. Based on Safra's Google Scholar profile, the authors have not done this work. But until this work has been done, any conclusions about trustworthiness based on the authors' methods are premature. In particular, attempting to replicate the headline finding of Safra et al. (2020) does nothing to support these crucial assumptions.

On the other hand, "the authors should have done a different study" is generally considered an unfair criticism, and the current project was a reasonable replication attempt of the headline finding of Safra et al. (2020). Whatever it is that the authors' technique measures, it was measured the same way as in Safra et al. (2020) and was correlated with time and GPD in Study 1, but not in Study 2.

I would be more supportive of publishing the current paper if

(1) the term "perceived trustworthiness" were replaced with a more hedged term, such as "estimated trustworthiness" or "modeled trustworthiness";

(2) the authors noted that there are substantial unanswered questions about the validity of their measurement technique;

(3) the paper were framed narrowly as a replication attempt of the headline finding of Safra et al. (2020), without attempting to draw any conclusions about trustworthiness or speculating about why the findings did not replicate. This would mean removing the discussion of modernization theory from the introduction, the references to social trust in §2.3, and most especially the claims about the scientific utility of the method in the second paragraph of the general discussion.

5. The article is presented in an intelligible fashion and is written in standard English.

Yes.

6. The research meets all applicable standards for the ethics of experimentation and research integrity.

Not human subjects or animal research.

7. The article adheres to appropriate reporting guidelines and community standards for data availability.

- An OSF link is provided in the metadata, but currently it's set to private/only visible by permission. The authors should make this link public before the paper is accepted.

- For Study 1, the authors provide two OSF links in the text, to two frozen archives of the project. The archives include the monarch data scraped from Wikipedia, but do not include any portrait image data, intermediate non-copyrighted data (such as extracted facial action units or "perceived trustworthiness" scores), or analysis scripts. Study 1 does not appear to be computationally reproducible given the contents of this repository.

- For Study 2, the authors provide an OSF links in the text, a frozen archive of a distinct project. The project is also currently set to private/only visible by permission. The archive includes the portrait/subject metadata for the portraits in the sample, but again does not include portrait image data, intermediate data, or analysis scripts. Study 2 also does not appear to be computationally reproducible given the contents of this repository.

7. PLOS authors have the option to publish the peer review history of their article (what does this mean?). If published, this will include your full peer review and any attached files.

Reviewer #1: No

Reviewer #2: No

---

## [Author Response · Author response to Decision Letter 1]

10 Jul 2023

All responses are in the cover letter.

---

## [Editor Report · Decision Letter 2]

26 Jul 2023

Using portraits to quantify the changes of generalized social trust in European history: A replication study

PONE-D-22-29440R2

Dear Dr. Baumard,

We’re pleased to inform you that your manuscript has been judged scientifically suitable for publication and will be formally accepted for publication once it meets all outstanding technical requirements.

Kind regards,

Anastassia Zabrodskaja, Ph.D.

Academic Editor

PLOS ONE
---

## [Editor Report · Acceptance letter]

6 Sep 2023

PONE-D-22-29440R2 

Using portraits to quantify the changes of generalized social trust in European history: A replication study 

Dear Dr. Baumard:

I'm pleased to inform you that your manuscript has been deemed suitable for publication in PLOS ONE. Congratulations! Your manuscript is now with our production department. 

Kind regards, 

on behalf of

Professor Anastassia Zabrodskaja 

Academic Editor

PLOS ONE